# Functional Dentition, Chronic Periodontal Disease and Frailty in Older Adults—A Narrative Review

**DOI:** 10.3390/ijerph20010502

**Published:** 2022-12-28

**Authors:** Gabriel Lee Keng Yan, Mei Na Tan, Mun Loke Wong, Chong Meng Tay, Patrick Finbarr Allen

**Affiliations:** 1Faculty of Dentistry, National University of Singapore, Singapore 119085, Singapore; 2National University Centre for Oral Health, Singapore 119085, Singapore; 3Department of Restorative Dentistry, Cork Dental School and Hospital, University College Cork, T12 DC4A Cork, Ireland

**Keywords:** dental status, oral function, gerodontology, frailty, functional aging, oral rehabilitation

## Abstract

Background: The likelihood of experiencing the impact of chronic disease rises with age, and cumulative functional deficits over time increase the risk of frailty in older adults. The exact causes of frailty are not clear, and research is needed to identify appropriate intervention measures to reduce risk of developing frailty in old age. Objective: To review the evidence on the relationship between frailty, dental status and chronic periodontitis and to determine if improvements in oral health of older adults can contribute to reversal of frailty. Results: The oral cavity is the entry point to the gastro-intestinal tract, and natural teeth facilitate efficient mastication of food prior to swallowing and subsequent digestion. The loss of natural teeth, which is gradual and cumulative over the life course, is associated with diminished nutritional intake, especially in older adults. Furthermore, chronic periodontitis has been postulated as a risk factor for frailty. The evidence supporting a strong relationship between oral health status and frailty is not clearcut. Cross sectional studies suggest an association with missing teeth and chronic periodontal inflammation. However, there are very few longitudinal studies and accordingly, it is not currently possible to claim a causal relationship. As yet, there is no evidence to suggest that improvements in oral health contribute to reversal of frailty. Conclusion: Longitudinal studies with robust designs are required to better inform the relationship across functional dentition, chronic periodontitis and frailty in older adults.

## 1. Introduction

There is a well reported global trend of aging and extended lifespan, and most people in economically advanced nations can now expect to live well beyond their sixth decade [1,2]. The likelihood of experiencing the impact of chronic disease rises with age, and the growing burden of healthcare in older people raises concerns about the sustainability of current models of healthcare.

There is a new focus on improving the healthy life span as the population ages, and in so doing, shift the focus from treatment of disease to prevention of disease in old age. Functional aging initiatives should not only address prevention but include the need to curtail the accumulated functional deficits that occur with chronic disease as people get older. Research is required to shed greater light on the interplay between factors which cause functional deficits, and effectiveness of interventions designed to reduce or reverse these deficits. This is particularly relevant to older adults where cumulative functional, metabolic and psychological deficits may lead to frailty and disability in later years of life.

The WHO [1] has articulated the need to take a life course approach to studying the epidemiology of disease across the generations, and to improve attempts at decelerating the rate of health decline as people age. This includes: targeting of health promotion interventions to reduce disease risk; specific interventions to reduce disability (including tooth loss), and identification of social inequalities likely to raise disease risks in specific populations. Ideally, all interventions would be designed to avoid people entering the disability threshold zone in older age. This means that risk reduction strategies for disease in old age must commence earlier in the life course to maximise impact.

The aims of this paper were to review the relationship between dental status, chronic periodontitis and frailty in older adults and to determine if improvements in oral health of older adults can contribute to reversal of frailty.

## 2. The Concept of a Functional Dentition

The oral cavity is the entry point to the gastro-intestinal tract, and natural healthy teeth are essential to efficient mastication of food prior to swallowing and subsequent digestion. The loss of natural teeth, which is gradual and cumulative over the life course, is associated with diminished nutritional intake, especially in older adults. Complete or partial dentures are used to replace lost natural teeth, but only partially or minimally attenuate the loss of chewing function following loss of natural teeth. The peak incidence for severe tooth loss is in the sixth decade of life [3]. The concept of “functional dentition” has been developed to describe the minimum number of natural teeth needed to provide adequate (as opposed to ideal) oral function. In studies of nutrition in adult populations, poor quality diet has been reported in adults missing natural teeth and wearing partial and complete replacement dentures [4,5]. The reasons for this are thought to be difficulty in chewing hard foods such as meats, raw vegetables and fruit and decreased sense of taste. There is also an association between hyposalivation (more common in older age), dietary intake, masticatory performance and swallowing [6,7,8]. Most commentators agree that a minimum of 20 natural teeth with 3–5 occluding pairs of posterior teeth with an intact anterior dentition are needed to provide a “functional dentition” to facilitate satisfactory (as opposed to “ideal”) chewing and aesthetic requirements [9,10]. The shortened dental arch (SDA) concept, initially proposed by Käyser and co-workers [11,12,13], recommends restoring tooth spaces anteriorly with fixed restorations, and where necessary, distal extension cantilevered fixed partial dentures to provide a minimum of three occluding pairs of posterior teeth. The intent of this strategy is not to provide a “complete” dentition, and removable partial dentures are not utilised. For this strategy to be successful, the remaining natural dentition should have a good long-term prognosis. This means that dental diseases are under control and the patient demonstrates a good standard of oral hygiene. A number of randomized clinical trials have shown that restoration to shortened dental arches in older adults have favourable outcomes, including improved chewing ability [14,15,16]. In Japan, recognizing the health benefits of a functional dentition, the health system adopted a target of a minimum of 20% of 80-year-old and 50% of 60-year-old citizens retaining a minimum of 20 natural teeth by 2010. This has been enshrined in the “Healthy Japan 21” strategy for health promotion in the 21st century [17]. However, the relationship between diet and dental status is confounded by food preferences, socioeconomic status, social circumstances and educational attainment. Restoration of missing teeth may improve objective chewing performance and bite force, but does not necessarily improve food choices without tailored dietary advice [18].

## 3. Relationship between Periodontitis and Systemic Health

A major cause of tooth loss in older adults is periodontitis, a disease that affects the supporting tissues that anchor the teeth to the jaws. It is caused by chronic inflammation in the periodontal tissues that develops in response to the bacterial challenge presented by the dysbiotic plaque biofilm that accumulates on the tooth surfaces [19]. If untreated, this progressive condition results in destruction of the periodontal ligaments and alveolar bone with increasing mobility of the teeth. Purulence and bad breadth are sometimes observed, and exposed dental root surfaces can become sensitive to thermal stimuli. The ultimate end point of untreated periodontitis is tooth loss. It is a highly prevalent noncommunicable disease (NCD), and advanced periodontitis is the sixth most prevalent NCD to affect mankind (prevalence 11.2%) [20]. There are established links between tooth loss, chronic periodontitis and a number of other chronic inflammatory diseases that affect older adults, in particular diabetes and cardiovascular diseases [21,22]. For example, there is evidence that patients with more than five missing teeth have increased risk for heart disease and myocardial infarction, and those with more than nine missing teeth have increased risk of diabetes and death from any cause [23]. In frail older adults, periodontal pathogens have been implicated in the aetiology of bacterial pneumonia with the potential for this to be life threatening [24,25].

With respect to the links between periodontitis and cardiovascular diseases, inflammation in the periodontal tissues leads to entry of bacteria into the blood stream, which activate host inflammatory responses by multiple mechanisms (elevated levels of C-reactive protein, altered endothelial function, increased production of acute phase proteins). The altered immune response favours atheroma formation and maturation, and there is consistent and strong epidemiological evidence that periodontitis imparts increased risk for future cardiovascular disease. Indeed, in a study of 805 adults under 75 years of age using case control design, the risk of a first myocardial infarction has been shown to be increased by the presence of chronic periodontitis, even adjusting for confounders [26].

With respect to type 2 diabetes mellitus, there is substantial evidence of a bi-directional relationship with periodontitis [27,28]. Untreated periodontitis results in increased levels of circulating bacteria and bacterial antigens, together with increased circulating levels of pro-inflammatory cytokines, such as interleukin-6 (IL-6), tumour necrosis factor-α (TNF-α) and oxygen radicals. The resultant upregulated systemic inflammatory state leads to impaired insulin signalling, and increased insulin resistance, leading to elevated HbA1c (glycated haemoglobin) levels [29]. In individuals who do not have diabetes, periodontitis is associated with higher HbA1c and prevalence of prediabetes (compared to individuals who are periodontally healthy), and severe periodontitis is associated with a significantly higher risk of developing diabetes (adjusted hazard ratio 1.19–1.33) [30,31]. In people with type 2 diabetes, periodontitis is associated with higher HbA1c levels and significantly worse diabetes complication than those who do not have periodontitis. It is also important to note the benefits that treatment of periodontitis has in improving glycaemic control. Nonsurgical management of periodontitis has been shown to reduce HbA1c levels in both prediabetic and diabetic patients [32]. Meta-analyses and Cochrane reviews have consistently demonstrated reductions in HbA1c of around 0.3–0.4% following periodontal treatment, a clinically relevant benefit equivalent to adding a second line pharmacologic agent to a diabetes treatment regime [33,34]. Accordingly, maintaining periodontal health in older age may help reduce extent and severity of major chronic illness and should be a priority.

## 4. Oral Health Status and Relationship to Frailty

In recent years, there has been growing interest in the study of aging and the potential to develop interventions which would delay the aging process and even prevent chronic disease. Kennedy et al. [35] proposed that age related chronic diseases are manifested once molecular, cellular and genetic factors converge with resultant chronic inflammation. Low-grade, chronic inflammation is recognised as a contributor to the aging process. Markers of biological aging have been proposed [36], and preventive therapy can be provided for those identified as having a negative biological aging profile. In a recent review paper, Baima et al. [37] discussed the role of periodontitis in a geroscience context, and how periodontitis could contribute to acceleration of the aging process. The interplay of mechanisms is complex, but research is needed to explore further how early management of periodontitis can attenuate ill health in old age. Frailty in older adults is defined as a “clinically recognizable state of increased vulnerability resulting from aging-associated decline in reserve and function across multiple physiologic systems such that the ability to cope with everyday or acute stressors is compromised” [38]. The aetiology of frailty is complex, with many proposed factors involved. These are summarized in the recently proposed model adapted from Castrejon-Perez et al. [39] shown in Figure 1. There is an interaction between age related physiological decline, chronic disease and socioeconomic determinants which contribute to gradual functional decline. Metabolic changes including decreased protein synthesis and increased production of free radicals, some age related and some chronic disease related, result in functional decline. At some point, the cumulation of these deficits is sufficient to cross the threshold of disability and ultimately, frailty.

There is a lack of consensus on characterization of frailty, but two models are most commonly used. The phenotype model proposed by Fried [40] uses five self-reported and objective indicators, namely: (1) recent weight loss; (2) exhaustion; (3) low physical activity; (4) muscle weakness, and; (5) slow walking speed. In the phenotype model, individuals are classified as “frail” if they manifest three or more of these indicators, as “prefrail” if they manifest one or two of the indicators, and as “nonfrail/robust” if they do not manifest any of the indicators. The second model was proposed by Rockwood and colleagues [41], and calculates a score based on the ratio of clinically measured cumulative deficits to a range of deficit possibilities. This is mapped onto a seven point clinical frailty scale to define the extent of frailty, ranging from fit/robust to severely frail [dependent/terminally ill].

Frailty is associated with increased falls risk, depression and reduced quality of life in old age and is a significant public health problem which is recognized as a distinct age-related condition [42,43,44]. There is a reported prevalence of 10% of adults over 65 years and 30% over 80 years of age [42]. Frailty and its sequelae are associated with increased healthcare costs, independent of age and co-morbidity [43]. The causes of frailty are unclear, and research is needed to shed further light on this and help identify appropriate intervention measures to reduce risk of developing frailty. Poor quality diet and low intake of dietary protein has been linked to development of frailty, but it is thought that frailty and malnutrition are related but distinct conditions. A recent study from Singapore (the Singapore Longitudinal Aging Study) showed that only 26% of the participants classed as “malnourished” were also classified as “frail” [44]. There was a lot of overlap in the risk factors for malnutrition, frailty and prefrailty. This finding is consistent with the conclusion of a recently published systematic review of the relationship between malnutrition and frailty [45], suggesting that timely intervention may have a positive impact on both nutrition and frailty status. It also suggests that studies that measure trajectory of frailty should also assess nutrition status. There have been a number of studies which assessed the impact of various interventions on frailty status. Most of the frailty interventions included in the systematic review [45] involved increased physical activity and other interventions involved health education, nutrition supplementation, home visits, hormone supplementation and counselling. A significant improvement of frailty status was demonstrated in 71% (*n* = 10) of studies and of frailty indicators in 69% (*n* = 22) of studies where measured. A study in Singapore using a parallel arm RCT design showed reversal of frailty states in 3–6 months, which was sustained at a 12-month assessment point [46].

In recent years, the concept of “oral frailty” has been proposed. How to define oral frailty is the subject of debate, but it is broadly agreed that the extent of oral frailty is related to dental status (number of occluding units), self-reported chewing ability, objective measurement of bite force and level of oral dryness. However, there is a lack of consensus on how to define oral frailty, with a recent scoping review reporting seven different definitions [47].

A question arises whether improving nutritional intake with the aid of improved oral function can reduce the risk of frailty. The relationship between oral health status and frailty has been the subject of a number of recent publications [48,49,50,51,52]. Dental disease frequently leads to loss of teeth, which in turn, compromises chewing function. In older adults, there may also be dysphagia or difficulties in swallowing, diminished motor control of oro-facial musculature and saliva gland hypofunction with associated dry mouth. Intuitively, anything which compromises adequate intake of nutrients such as proteins is potentially linked to onset of frailty. However, the lack of longitudinal data means that there is not necessarily a cause and effect link between dental status and frailty. Studies in Japan have indicated an association between chewing ability and sarcopenia (defined as loss of skeletal muscle mass) in elderly Japanese community dwelling adults and onset of dementia. These studies were cross sectional in nature, and determined that the number of remaining teeth, occlusal contacts and bite force were all related to food intake and cognitive function. In the study reported by Tanaka et al. [52], the authors proposed “oral frailty” as having poor markers in three or more of six conditions: (1) number of remaining teeth; (2) chewing performance; (3) tongue mobility; (4) oral motor skills; (5) tongue pressure and (6) self-reported chewing ability. A total of 16% of their sample of 2000 Japanese adults over 65 years of age were reported as having “oral frailty”. In multivariate analysis, having oral frailty doubled the risk of having sarcopenia, frailty, disability and mortality. Great care needs to be exercised in interpreting results of cross sectional studies, and causality should not be assumed. A recent longitudinal study with two-year follow-up reported that “oral frailty” doubled the risk of deterioration in nutrition status in older Japanese adults [53]. The authors acknowledged some limitations in this research, including the lack of detail on dental treatment and changes in social supports between baseline and follow up. Further work is required to investigate this relationship in different contexts and with longer follow up periods.

Recently, there have been studies of biomarkers of aging and the possibility of determining the pace of individual aging in younger adults prior to age-related chronic disease experience. One such study included periodontal health status as one of the biomarkers, and reported reduced physical and cognitive health in 38-year-old adults with older “biological” age [36]. It is noteworthy that evidence is emerging to support the hypothesis that chronic systemic inflammation may contribute to the development of frailty [54]. Circulating levels of IL-6, CRP and TNF-α are among those pro-inflammatory cytokines that have been independently associated with frailty. In a recent systemic review and meta-analysis which included nearly 24,000 subjects in 32 cross sectional studies, there was a large difference in circulating IL-6 and CRP in frail and prefrail subjects compared to robust subjects [55]. This relationship is likely to be moderated by age, elevated BMI/obesity and co-existing chronic disease states, and this needs to be analysed further in longitudinal studies. Currently, there are very few longitudinal studies of incident frailty and its association with chronic inflammation. A 10-year follow up study in the UK reported that participants with higher baseline white cell count and elevated cortisol levels were more likely to become frail 10 years later [56]. However, the systematic review referred to earlier [55] could only identify this study and two others for inclusion in their review. They were not able to demonstrate an association between inflammatory markers and incident frailty. The authors offered some observations, including the relative paucity of longitudinal data, the possibility that their meta-analysis over adjusted for confounders, and the lack of consistency in the definition of frailty used in these studies. Given that older adults are more prone to sub-acute infections causing an elevation of biomarkers of inflammation at the time of data collection, this potential limitation should not be ignored. Nevertheless, the data reported support the ongoing investigation of the possible relationship between chronic inflammation and frailty.

Periodontal disease is mostly chronic in nature, and has a high prevalence in older adults. Immune senescence, which refers to the process of altered immune functioning that occurs with increasing age is also considered to result in increased susceptibility of older individuals to infections. A systematic review of immune senescence and periodontitis identified evidence for altered neutrophil function and increased production of pro-inflammatory mediators (e.g., IL-1β, IL-6) in older compared to younger subjects, and animal experiments suggested increased expression of genes that contribute to a pro-inflammatory state in older compared to younger animals [57]. There is clear evidence, therefore, that immune functioning related to periodontitis alters with increasing age, and the systemic impacts of this remain to be more clearly elucidated.

Currently, the relationship between periodontitis and frailty is poorly understood. However, it is plausible that periodontitis (which is increasingly prevalent in older populations and which causes upregulated systemic inflammation) may contribute to frailty, not only as a result of functional impacts (e.g., associated with tooth loss and impacts on nutritional status), but also via increased inflammation. Lee et al. [58] reported that frailty states can deteriorate and improve over time, and the direction of change is related to the management of conditions such as stroke and cognitive impairment. They highlighted that diabetes is of particular concern, as it can result in muscle wastage in older adults. Given the reported relationship between periodontal disease and diabetes, it may be possible to reverse frailty states with timely interventions designed to reduce periodontal inflammation. However, as yet, the contribution of improved oral health toward this is not understood. The onset of frailty in older age coincides with the severest cumulative effects of chronic periodontitis manifested by periodontal inflammation, tooth mobility, decay of exposed root surfaces and ultimately, tooth loss. Castrejón-Pérez et al. [39] analysed the relationship between three-year incident frailty and oral health and reported that risk ratio of incident frailty was double for adults with severe periodontitis. Each additional tooth reduced the probability of frailty by 5%. However, the limitations to this study were that they did not measure incident periodontal disease and there was a small sample size. The main limitations of currently published studies in respect of links between periodontitis and frailty, which are conflicting and inconclusive, include: (a) lack of longitudinal studies; (b) failure to include important covariates in the analyses, particularly, baseline cognitive function and medication status; (c) lack of inflammatory biomarkers; (d) use of crude measures to record periodontal status; and (e) lack of suitable control participants.

## 5. Conclusions

In conclusion, tooth loss does impact chewing function and may compromise the nutritional value of one’s diet when the residual dentition is not functional. Poor diet is not a direct consequence of a reduced dentition, but it may be a contributing factor for frailty in old age. Accompanied by tailored diet advice, maintaining a functional dentition, perhaps with the aid of a dental prosthesis, may facilitate improved diet. Periodontal disease is implicated as a complicating factor in many disease states, including diabetes mellitus. If these conditions are not well managed, then they may contribute to the development of frailty. If periodontal health can be stabilized in patients with diabetes and a frailty risk, this may reduce the risk of frailty. The evidence to support this is not yet robust, and further longitudinal studies are required to determine if improved periodontal health reduces the risk of frailty in older adults.

## Figures and Tables

**Figure 1 ijerph-20-00502-f001:**
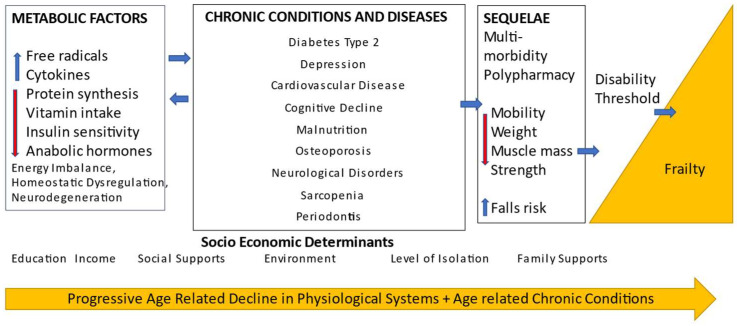
Factors contributing to the incidence of frailty [Adapted with permission from Ref. [39]. Copyright 2017 Oxford University Press].

## Data Availability

Not applicable.

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
