# Peer review of "Functional Dentition, Chronic Periodontal Disease and Frailty in Older Adults—A Narrative Review"

_ijerph, 2022, doi:10.3390/ijerph20010502_

Round 1

Reviewer 1 Report

The aim of this project titled “Functional Dentition, Chronic Periodontal Disease and Frailty in Older Adults” was to review the relationship between dental status, periodontal disease, and frailty in elderly and to determine if improvements in oral health of older adults can contribute to reversal of frailty. 

Minor comments:

The manuscript overall needs minor editing of its English language and style/flow of phrases used, but overall is reads very well.

P4-Line 72-73 “Furthermore, missing posterior teeth are not replaced with a prosthesis when 3-5 occluding pairs of natural teeth are present.” Can you further explain and add appropriate citations?

In few instances the authors use the term periodontal disease interchangeably with periodontitis which is not accurate. What applies for periodontitis, especially in the realms of systemic conditions, does not apply to gingivitis for example. Please change accordingly.

Does the authors anticipate they will be adding periodontitis to Figure 1 since this is the topic of the current manuscript.

Although I like the flow of topics discussed, the direct association of periodontitis with frailty in the elderly population is only discussed/justified in the last few paragraphs. I suggest the authors elaborate on this since it the main matter of interest.  

Reviewer 2 Report

This report provides a narrative summary of the literature suggesting that edentulism and periodontitis can contribute to frailty in older adults. Although many reviews have been already published, there is substantial value in exploring this concept. However, I do have a number of comments that should be addressed in the manuscript as follows:

- Please, add a reference for the definition of periodontitis from the AAP/EFP world workshop (doi: 10.1002/JPER.17-0721)

- Considering the multi-morbid state of the majority of the elders, it would be relevant to underline that the relationship between periodontitis and type 2 diabetes is of bidirectional nature (doi: 10.3390/jcm10081787)

- Pag 10 line 206. Indeed, some longitudinal reports do exist correlating systemic marker of low grade chronic inflammation with accelerated aging and incipient frailty (doi: 10.1073/pnas.1506264112)

- A bunch of recent investigations have been linking periodontitis to accelerated biological aging and cellular senescence (for a review see doi: 10.1177/00220345211037977). This work may provide additional insight on the complex network lying behind the relationship between oral health, aging and NCDs.

- Pag 12 lines 234- when talking about the effect of periodontal treatment on the longitudinal metabolic health, the seminal work of Kocher et al could not be neglected (doi: 10.1177/0022034518804185)

In general, the review reads well, although additional effort should be made to expand its sections based on my previous comments and a more thorough literature search. 

Round 2

Reviewer 2 Report

The authors have addressed all changes and the manuscript has been improved except for some grammar and style errors.